# Analysis of the Entire Mitogenome of the Threatened Freshwater Stingray *Potamotrygon leopoldi* (Myliobatiformes: Potamotrygonidae) and Comprehensive Phylogenetic Assessment in the Xingu River, Brazilian Amazon

**DOI:** 10.3390/ijms26178252

**Published:** 2025-08-26

**Authors:** Sávio L. M. Guerreiro, Amanda F. Vidal, Caio S. Silva, Giovanna C. Cavalcante, Leandro Magalhães, Daniel H. F. Gomes, Júlio César da Silva Filho, Jorge E. S. de Souza, Éder Pires, Guilherme Oliveira, Debora Sayumi Doami Melo, André Luiz Alves de Sá, Igor Hamoy, Ândrea Ribeiro-dos-Santos, Sidney E. B. Santos

**Affiliations:** 1Laboratory of Human and Medical Genetics, Graduate Program in Genetics and Molecular Biology, Federal University of Pará, Belém 66075-110, Brazil; scaio@hotmail.com (C.S.S.); giovannaccavalcante@gmail.com (G.C.C.); akelyufpa@gmail.com (Â.R.-d.-S.); sidneysantosufpa@gmail.com (S.E.B.S.); 2Instituto Tecnológico Vale (ITV), Belém 66055-090, Brazil; amandaferreiravidal@gmail.com (A.F.V.); leandromag@me.com (L.M.); ederpires@yahoo.com.br (É.P.); guilherme.oliveira@itv.org (G.O.); 3Bioinformatics Multidisciplinary Environment (BioME), Digital Metropolis Institute, Universidade Federal do Rio Grande do Norte (UFRN), Natal 59078-900, Brazil; daniel.gomes.702@ufrn.edu.br (D.H.F.G.); juliocesarfilho0112@gmail.com (J.C.d.S.F.); jorge@imd.ufrn.br (J.E.S.d.S.); 4Laboratório de Genética Aplicada, Programa de Pós-Graduação em Saúde e Produção Animal, Universidade Federal Rural da Amazônia (UFRA), Belém 66077-830, Brazil; doami2211@gmail.com (D.S.D.M.); sa.andrealves@gmail.com (A.L.A.d.S.); ighamoy@gmail.com (I.H.)

**Keywords:** mitochondrial genome, positive selection, phylogeny, arraia Negra do Xingu

## Abstract

The present study characterizes the complete mitochondrial genome of *Potamotrygon leopoldi*, commonly referred to as the “white blotched stingray”, a member of the Potamotrygonidae family that are a group of stingrays that live exclusively in freshwater environments. *Potamotrygon leopoldi*, endemic to the Xingu River in the Amazon region, are exploited by commercial fisheries for food and commonly exploited by the ornamental industry, and this has a significant impact on the populations. Here, newly assembled PacBio long-read sequencing assesses the complete mitogenome of *P. leopoldi* and performs a comparative investigation into the evolutionary connections to other extant taxa of elasmobranchs. The mitogenome has 17,504 bp, containing 13 protein-coding, 22 tRNA, and 2 rRNA genes. The mitogenome comprises A: 32.32%, T: 24.41%, C: 12.84%, and G: 30.42%, with an AT content of 56.73%. The values of AT and GC skewness were 0.13 and −0.40, respectively. Our phylogenetic analyses with mitogenome sequences of 40 elasmobranch species support the monophyly for the Potamotrygonidae family and indicate a close relationship to the Dasyatidae family and a sister relationship with *Potamotrygon orbignyi* and *Potamotrygon falkneri*. We also detected various amino acid sites in positive selection exclusively in *P. leopoldi*. This extensive comparative mitogenomic investigation offers novel and significant insights into the evolutionary lineage of neotropical freshwater stingrays and their closely related taxa. It is an indispensable resource for facilitating ongoing and prospective investigations into the molecular evolution of elasmobranchs.

## 1. Introduction

Mitochondrial DNA (mtDNA) exhibits a relatively high mutation rate compared to nuclear DNA, largely due to its exposure to reactive oxygen species and limited DNA repair mechanisms. These mutations are subject to selective pressures, particularly purifying selection, which tends to remove harmful nonsynonymous mutations that could disrupt essential mitochondrial functions like ATP production. In contrast, synonymous mutations, which do not alter protein sequences, may accumulate more freely, although they can also be subject to positive selection if they influence gene expression or mRNA stability [1].

The mitochondrial DNA (mtDNA) has about ten to twenty times the nuclear DNA (nDNA) mutation rate, partially because of the absence of efficient mtDNA repair systems [2]. Due to this mutation rate, the mitochondrial genome is an excellent tool to compare with other species and to help understand the recent evolutionary events that the animals suffered [3].

Stingray species have a fundamental ecological role in aquatic ecosystems, but many are usually fished for human consumption and the aquarium industry. Data displayed by the Ministry of Development, Trade, and Services of Brazil show that the trade of ornamental freshwater fishes accounted for BRL 2.3 million in 2023 [4]. Among these species, Potamotrygonidae stingrays are sought after for domestic and international markets due to their conspicuous coloration patterns. Potamotrygonidae is a monophyletic family of stingrays that evolved to live exclusively in freshwater environments and includes four genera: *Potamotrygon*, *Paratrygon, Heliotrygon*, and *Plesiotrygon* [5,6,7]. 

The genus *Potamotrygon* exists in South America, with 39 species described, including species with wide distribution (e.g., *P. motoro* and *P. orbignyi*) and endemic species restricted to specific rivers (e.g., *P. leopoldi* and *P. henlei*) [8]. Notably, *P. leopoldi* inhabits the downstream waters of the Xingu River in the Amazon region and is largely used as an ornamental fish, making it particularly vulnerable to extinction. The species is listed in Appendix III of CITES, classified as Vulnerable on the IUCN Red List, and subject to quotas for occasional capture for export purposes, regulated by IBAMA in Brazil [9,10,11,12]. *P. leopoldi* is distinct within its genus due to its unique black coloration with white spots and broader disk width, facilitating its ease of identification compared to related species like *P. falkneri* and *P. motoro*, which have subtler patterns. It is also differentiated from *P. henlei* by its smaller, more numerous teeth (23–25 rows) and its distribution in the Xingu and Iriri rivers, as opposed to the Tocantins and Araguaia rivers where *P. henlei* is found. These morphological distinctions are essential for accurate species identification in ecological research [9]. Although there has been an increase in the knowledge about the biogeography and evolution of freshwater stingrays [13], there are still few studies that focus on the molecular changes associated with the emergence of adaptive characteristics allowed for the adaptation of the stingrays from a marine environment to a freshwater ecosystem. In this sense, a comprehensive analysis of the mitochondrial genome may be a valuable strategy to expand our understanding of the evolutionary history of these species [14]. Currently, the mitogenomes of five species have been described for the *Potamotrygon* genus, and reinforcements in using genomics are necessary to study freshwater stingrays.

Mitochondrial genomes have become a powerful tool for enhancing species identification and guiding conservation strategies, particularly for closely related species where single-gene barcoding methods like COI are insufficient. The mitogenome includes various mitochondrial markers that improve the resolution of species identification and allow for more detailed population genetic studies. This comprehensive genetic information is particularly valuable in eDNA research that uses mini barcodes, where these additional markers help enhance biodiversity assessments and provide a complete reference database. For species like *P. leopoldi*, which is vulnerable to extinction, the mitogenome offers a critical resource for monitoring genetic diversity, identifying distinct population units, and informing management strategies to ensure long-term survival. By linking evolutionary history with current conservation efforts, the mitogenome enhances the effectiveness of biodiversity monitoring and population management initiatives [15,16].

Therefore, we report the assembly and annotation of the complete mitochondrial genome of *P. leopoldi* and use the publicly available mitochondrial mitogenome of the Myliobatiformes to investigate the evolutionary dynamics compared with the marine group. Our investigation focused on understanding the evolutionary dynamics of *Potamotrygon* mitogenomes by comparing them with other stingrays and elasmobranchs to understand the differences and the molecular adaptation of the freshwater stingrays. Additionally, we analyzed gene synteny to identify potential rearrangements within the Potamotrygonidae family, and we tested for amino acid sites under positive selection in the mitochondrial genes to search for insights into the molecular adaptation of *P. leopoldi* and *Potamotrygon* genus. A mitogenome phylogeny was generated using data from the mitogenomes of rays and sharks, enabling us to infer the phylogenetic relationship with the *Potamotrygon* genus, and the evolutionary time of speciation.

## 2. Result

### 2.1. Mitochondrial Genome Structure

The complete circular mitogenome of *P. leopoldi* was sequenced using a Sequel I platform by PacBio Systems, generating a total of 6,325,249,036 reads. We also used a set of sequences generated by Zhou et al., 2023 [17], with the Illumina Hiseq platform. It should be noted that Zhou et al., 2022 [17] assembled the total nuclear genome of such specimens but the mitochondrial genome remained unannotated and with a large amount of redundancy. After these analyses, the total length of the mitogenome was 17,504 bp, containing 13 PCGs, 22 tRNAs, and 2 rRNA genes (Figure 1). The *P. leopoldi* mitogenome was deposited in Genbank under accession number OR896919. 

The mitogenome of the white blotched stingray has a nucleotide composition as follows: A: 32.32%, T: 24.41%, C: 12.84%, and G: 30.42%. The A+T content is higher than the C+G content, showing that the mitogenome is biased towards AT. The AT and GC skewness of the mitogenome sequence was 0.13 and −0.40, showing the A-skew and C-skew. Regarding the genes, 28 are encoded in the H-strand (Heavy strand, +), and the other nine are encoded in the L-strand (Light strand, −), as seen in Table 1. The arrangement of genes was typical for vertebrates except for the variation in the length and the intergenic spacer region.

The length of tRNA genes varied from 67 bp (tRNASer) to 75 bp (tRNALeu). All tRNA presented a typical cloverleaf secondary structure with a few base pair mismatches (Figure 2). The nucleotide frequencies of rRNA are A: 35.63%, T: 21.12%, G: 17.27%, and C: 25.98%, with an AT content of 56.75%. Ten tRNAs showed secondary structure with the following mismatches: G-G base pair (instead of G-C), T-T (instead of T-A), A-A (instead of A-T), and G-T (instead of G-C) (Figure 2).

The gene order and orientation, along with the distribution of genes on both the heavy and light strands, exhibited complete congruence with the mitogenomes of other representatives within the *Potamotrygon* genus, including *P. falkneri, P. magadlenae, P. motoro, P. orbignyi*, and the outgroup species, *C. milii*. We illustrate this notable similarity among the species in Figure 3. These findings underscore the remarkable conservation of gene organization and strand distribution within the *Potamotrygon* genus and contribute to our understanding of their evolutionary relationships.

### 2.2. Protein-Coding Genes (PCGs)

The length of the protein-coding genes within the mitogenome of *P. leopoldi* was determined to be 11,444 base pairs (bp), with individual gene lengths ranging from 168 bp (ATP8) to 1836 bp (ND5). The overall average A+T content was calculated to be 57.0%.

Comparative analysis with other Potamotrygonidae species revealed a significant similarity in using conventional start and stop codons within most of the PCGs. Specifically, the majority of PCGs in *P. leopoldi* start the translation with the start codon ATG, and they commonly terminate with the stop codon TAA. However, notable exceptions were observed in two genes: the COXI gene utilized the start codon GTG, diverging from the typical ATG start codon pattern, and the ND6 gene employed the stop codon TAG instead of the more common TAA stop codon, similar to that noted in most fish species [18].

The PCGs comprised approximately 64.70% of the total mitochondrial genome in *P. leopoldi*. The average effective number of codons, an indicator of codon usage bias, was determined to be 57.65. Our analysis predicted a total of 3811 codons from the 13 PCGs found. Figure 4 displays the Relative Synonymous Codon Usage (RSCU) for amino acids, revealing a significant bias in codon usage for most of them. Comparative analysis of RSCU across mitogenomes from various elasmobranch families demonstrated interesting similarities. When comparing the five representatives of the Potamotrygonidae family with the outgroup *C. milii*, the codon usage patterns showed consistency, further supporting the close phylogenetic relationships among these taxa. These comprehensive results provide valuable insights into the genomic features and codon preferences within the mitogenome of *P. leopoldi* and contribute to improving our understanding of codon usage patterns and evolutionary relationships among elasmobranch families.

### 2.3. Control Region

The mitochondrial genome of *P. leopoldi* shares a conserved arrangement observed in other *Potamotrygon* mitogenomes, characterized by the positioning of the control region (CR) between the tRNAPhe and the tRNAPro (as illustrated in Figure 1). Notably, in *P. leopoldi*, this CR exhibits a high abundance of adenine (A) and thymine (T), resulting in a base composition of A = 33.55%, T = 29.92%, C = 24.30%, and G = 12.24%.

Within the CR of *P. leopoldi*, our investigation revealed the presence of two types of tandem repeats, the first consisting of a larger 128 bp-long repeat and a smaller one with 20 bp (Figure 5). These findings showed the genomic organization and sequence characteristics of the *P. leopoldi* mitogenome, offering valuable insights into the genetic diversity and evolutionary aspects of this species.

### 2.4. Adaptive Molecular Evolution

In the mitogenome analysis, we found sites in positive selection when compared with Myliobatiformes that have mitogenomes annotated in GenBank: ND1-1 sites under positive selection, ND4-1 sites under positive selection, ND5-3 sites under positive selection, COX I-6 sites under positive selection, COX II-3 sites under positive selection. With this analysis, we observe that the positive selection may be related to the functional adaptive adjustment to the freshwater stingrays that have the adaptation to survive and colonize the freshwater in the Amazon Basin.

### 2.5. Phylogenetic Analyses 

In the phylogenetic tree created using the BI method, the species display strong nodal support, forming separate clades for different families. Particularly, the family Potamotrygonidae is observed to be a monophyletic group that shares a sister relationship with the Dasyatidae family. Additionally, the Myliobatidae family is positioned as a basal group. Further examination of the tree reveals a sister relationship between *P. leopoldi*, *P. falkneri,* and *P. orbignyi* (Figure 6).

## 3. Discussion

### 3.1. Mitochondrial Genome Characterization

This article provides a comprehensive sequencing of the complete mitochondrial genome of *P. leopoldi*. The sequencing approach yielded a coverage of 11×, and we complemented our data with publicly available sequences previously published by Zhou et al. in 2022 [17]. The utilization of next-generation sequencing (NGS) technology in this study presents multiple advantages, notably the generation of long reads, which helps compensate for the relatively lower coverage achieved. Furthermore, the incorporation of short-read sequences contributes to enhancing the assembly process, leading to a more precise description of the mitochondrial genome [19].

Within the Potamotrygonidae family, the length of the mitogenomes varies, traditionally ranging from 17,429 bp (*P. magdalenae*) to 17,449 bp (*P. orbignyi*) [20]. Our analysis revealed that the mitogenome of *P. leopoldi* presents a length of 17,504 bp, making it larger than that of *P. orbignyi*, which previously held the record as the largest mitogenome described within the Potamotrygonidae family, with 17,449 bp [20].

This variation in mitogenome size can be attributed to differences in the length of the control region and intergenic regions. These regions are known to experience less influence from selection and exhibit a higher mutation rate, contributing to substantial variation among species [21]. Our study provides important insights into the genomic diversity within the Potamotrygonidae family and highlights its significance in shaping mitogenome size variation.

In the mitochondrial genome, nucleotides exhibit differential distribution between the two DNA strands, giving rise to DNA asymmetry [22]. To assess this asymmetry in the studied mitogenome, AT/GC skewness values were calculated. In the mitogenome of *P. leopoldi*, there are observed positive values for AT skewness, indicating an abundance of adenine (A) over thymine (T). Conversely, negative values were noted for GC skewness, implying an excess of cytosine (C) relative to guanine (G). The observed asymmetry can be attributed to the outcomes of mitochondrial replication and transcription, as these molecular processes result in the unwinding of the DNA, thereby facilitating the transient exposure of single-stranded DNA regions that exhibit an elevated susceptibility to mutational events [23,24,25]. The AT and GC skew values observed in the mitogenome of *P. leopoldi* were similar to those detected in other *Potamotrygon* species, such as *P. falkneri* [26]. These contribute to a better understanding of the genomic characteristics and evolutionary aspects within the *Potamotrygon* genus.

### 3.2. Protein-Coding Genes

The gene arrangement within the mitochondrial genome of *P. leopoldi* exhibited notable similarity to the commonly reported arrangement observed in most vertebrates [27]. Through a comprehensive comparative analysis involving four mitogenomes of species from the *Potamotrygon* genus, there are found a remarkably conserved gene arrangement, even in comparison to the outgroup species *C. milii* (Figure 3). Collectively, these findings strongly indicate a highly conserved mitogenome organization within the elasmobranchs, especially in the Potamotrygonidae family [20,28,29]. Nachtigall, Loboda and Pinhal [26] suggested that rearrangements in the mitochondrial genome of Batoidea species may be uncommon, which is in line with our findings that Potamotrygonidae mitogenomes have a conserved arrangement compared to other vertebrates. These observations contribute valuable insights into the genomic organization and evolutionary patterns within the Potamotrygonidae family and highlight the significance of conserved gene arrangements in these species.

In many species, synonymous codons are not used at an equal frequency for specific genes, resulting in codon usage bias. Previous studies have reported the existence of such bias in the mitochondrial genomes of elasmobranchs and other fishes [30]. There are observed codon usage biases for several genes among the species within the Potamotrygonidae family. This occurrence can be influenced by multiple factors, such as gene length, translational selection, tRNA abundance, mutation rates, GC composition, and the amino acid composition of proteins.

To quantify the degree of codon bias within and among species, the effective number of codons (ENc) is employed. This metric ranges from 20 to 61, where a value close to 20 indicates that only one codon from each codon family is used, while a value close to 61 indicates that all synonymous codons are utilized equally [31]. In our investigation, *P. leopoldi* exhibited an elevated ENc value of 57.65, meaning that synonymous codons are equally used within the Potamotrygonidae family. This observation aligns with the findings depicted in Figure 4. These results provide valuable insights into the codon usage patterns within the Potamotrygonidae family and enhance our comprehension of the factors influencing codon bias in mitochondrial genomes of freshwater stingrays.

#### 3.2.1. Control Region (CR)

As in other Myliobatiformes mitogenomes, the CR found here has tandem repeat regions with variable copy numbers and lengths of repeat units (Figure 5). Our analysis showed that the CR of *P. leopoldi* has a similar CR pattern (Figure 5) to other representatives in the *Potamotrygon* genus that have information on complete mitogenome in the public databases. Variability in the tandem repeat regions is common in elasmobranchs, and the high content of tandem repeats is responsible for the similar mtDNA genome size among the elasmobranchs [32]. Generally, the published mitogenomes of Potamotrygonidae lack discussion regarding such sequences within the control region. These features could potentially play a role in the regulation of replication in the mitochondrial genome, as suggested in other vertebrates (e.g., [28,33]).

The noncoding regions of the mitochondrial genome of *P. leopoldi* reveal significant insights into its evolutionary dynamics and ecological adaptations. Notably, the origin of light strand replication (OL) is located between the tRNA_Asn_ and tRNA_Cys_ genes, with sizes varying between species, while the control region (D-loop), positioned between tRNA_Pro_ and tRNA_Phe_, exhibits extensive size variation and is enriched in tandem repeat sequences. This tandem repeat variability in the control region, a common feature in elasmobranchs, may play a crucial role in regulating mitochondrial DNA replication and transcription, which is essential for maintaining genetic stability under environmental pressures. The observed polymorphism in nucleotide and size of the control region not only underscores the adaptability of *P. leopoldi* in freshwater habitats but also offers valuable insights into its population history and potential adaptive strategies, reflecting broader evolutionary trends within the *Potamotrygon* genus [28].

#### 3.2.2. Adaptive Molecular Evolution

Based on the results obtained in the study, positive selection was found in ND1, ND4, ND5, COX I, and COX II. It demonstrates that the *P. leopoldi*, when compared with other Myliobatiformes and *Potamotrygon* species, has supported the hypothesis of the migration of the marine stingrays millions of years ago with an adaptation, in this way, can see the regions in the positive selection that characterize the signature in the mitogenome. Supporting this idea, several studies have shown that, in ND genes, there is a functional relation to the proton piping process in the oxidative metabolism and are under strong positive selection in vertebrate species [34,35,36,37,38,39].

Noll et al. [39], found positive selection in penguins, and the complex ND1 and ATP complex, play a central role in cellular energy production and are the genes present in the greatest evidence in selection in marine organisms. A study by Sebastian et al. [40] with clupeoid fish that colonized marine habitats over millions of years presented positive selection in PCGs of animal lineages that migrate from marine to freshwater in genes ND2, ND4, ND5, and ND6 and has a relatively high number of amino acid changes. In the freshwater stingrays, the selective pressures are very similar to those in the clupeoid fish and would result from the selective pressures coming from niche changes, such as the transition from saltwater to freshwater environment. Indeed, when conducting molecular evolutionary analysis on the clupeoid, certain mitochondrial genes have been identified with sites subject to positive selection in freshwater species, as opposed to their closely related saltwater counterparts [35].

Sites under positive selection in the mitochondrial genome of *P. leopoldi* exemplify how species adapt to environmental changes, particularly in freshwater ecosystems. Similarly, a study on the elasmobranch *P. falkneri* found comparable patterns of positive selection in mitochondrial genes related to energy metabolism, indicating adaptations to dynamic environments with variable oxygen and ion concentrations. Both studies highlight the importance of mitochondrial evolution in driving species-specific adaptations to ecological niches, emphasizing how environmental pressures shape genetic modifications. These adaptations are critical for species resilience, particularly amid habitat changes due to anthropogenic impacts and climate change [26].

### 3.3. Phylogenetic Analyses

The phylogenetic analyses in this study illustrate the evolutionary position of the *Potamotrygon* species (Figure 6). Our results reveal a close relationship between *P. leopoldi* and other species from the Potamotrygonidae family. The sequence of the five representatives has a high similarity. The tree demonstrates monophyly, supporting the findings of Nachtigall, Loboda, and Pinhal [26], who inferred the phylogeny of the Potamotrygonidae family and proposed that the genus *Potamotrygon* is also monophyletic. Additionally, gene similarity comparisons indicate a strong resemblance among species within the *Potamotrygon* genus, particularly in the mitochondrial sequence. This similarity is likely due to the well-established evolutionary biogeographic history of the genus.

## 4. Materials and Methods

### 4.1. Sample Collection and DNA Extraction

Tissue samples were collected from a captive population of stingrays (*P. leopoldi*), two males and two females, at a farm that captured wild individuals to an exportation facility located in Salinópolis, Pará, Brazil. The four specimens collected were identified to the species level by author SLMG from this study using the key to the identification of the species of the genus *Potamotrygon*. The necessary permissions and access to the samples were obtained through the Secretary of Environment and Sustainability from Pará State (SEMAS-PA) under the Cooperation Agreement Nº05/2018.

Total genomic DNA was extracted from small fragments of the pectoral fin of adult *P. leopoldi* specimens using the Wizard Genomic DNA Purification Kit extraction method (Promega, Madison, WI, USA) in strict accordance with the supplier’s instructions. Subsequently, the concentration and quality of the extracted DNA were assessed using both NanoDrop 1000 spectrophotometer and Qubit 2.0 fluorometer (Thermo Fisher Scientific, Waltham, MA, USA). These procedures ensure the reliability and validity of the genomic DNA samples obtained for further analysis.

### 4.2. Assembly, Annotation, and Bioinformatic Analysis of the Complete Mitogenome

Two genomic libraries were prepared (one consisting of two samples from a male and the other of two samples from a female) utilizing the SMRTbell Template kit 1.0 (Pacific Biosystems, Menlo Park, CA, USA) following the manufacturer’s recommended protocols. The subsequent sequencing of these libraries was conducted on a PacBio Sequel I system (Pacific Biosystems, Menlo Park, CA, USA) utilizing an SMRT Cell 1M v3 (Pacific Biosystems, Menlo Park, CA, USA). In addition to our generated data, there are incorporated a set of complementary sequences of the *P. leopoldi* genome (CRR191193, CRR191177, CRR191191, CRR191189, and CRR191192), recently generated by Zhou et al. 2023 [17] employing the Illumina HiSeq platform. This integration of complementary data sources contributes to a more comprehensive, accurate and robust analysis of the assembly in our study, using strategies of hybrid assembly using two technologies of NGS. 

Raw sequencing data were filtered to trim adapters and low-quality sequences using CANU software version 2.2 [40]. The resulting reads were then assembled using Unicycler [41]. Protein-coding genes (PCGs), ribosomal RNAs (rRNAs), and transfer RNAs (tRNAs) were predicted by using MitoAnnotator [42] and annotated by alignment with other mitogenomes of *Potamotrygon*. The secondary structure of tRNAs was visualized in the Forna web server http://rna.tbi.univie.ac.at/forna/ (accessed on 2 December 2024) [43]. The tandem repeat units of the Control Region (CR) were identified with the Tandem Repeats Finder Server (http://tandem.bu.edu/trf/trf.advanced.submit.html (accessed on 2 December 2024); with default settings, except for “Minimum Alignment Score To Report Repeat” = 30) [44]. The AT and GC skews were calculated according to the following formulas: AT skew = (A − T)/(A  +  T) and GC skew = (G − C)/(C  +  G) [45]. Arrangement of genes encoding rRNA proteins, comparative relative synonymous codon usage (RSCU), and CR organization analysis were performed using in-house Python scripts. For these analyses, the following mitogenomes deposited in Genbank: *Potamotrygon falkneri* (MZ203140), *Potamotrygon orbignyi* (MN178254), *Potamotrygon motoro* (KF709642), *Potamotrygon magdalenae* (KX151183) and *Callorhinchus milii* (HM137147).

### 4.3. Phylogenetic Analysis

We reconstructed the phylogenetic tree using the complete mitogenome currently available for 40 species of elasmobranch [26,29,46,47,48,49,50,51,52,53,54,55,56,57,58,59,60,61,62,63,64,65,66,67,68,69,70,71,72,73,74,75,76,77]. The species *C. milii* (HM137147) was used as an outgroup. To reconstruct the phylogenetic tree, we performed multiple sequence alignment using the ClustalW alignment program with the default parameters in MEGA11 [78]. Bayesian Inferences (BI) were run using Markov-Chain Monte Carlo searches on two simultaneous runs of chains of 5,000,000 generations in Mr. Bayes version 3.1.2, with a sample frequency of every 1000th [79]. The first thousand trees were discarded as burn-in, and the posterior probability of each node was calculated from the remaining trees and examined in Figtree v1.4.4 [80]. 

The divergence time was estimated with BEAST v2.6.6 [81], and the reference in a million years (MYA) between genera *Pristis* and *Rhynchobatus* (67.8–159.0 MYA) [82] was used as prior (www.timetree.org) for calibration [83]. 

To measure the effective number of codons (ENc) is estimated through DAMBE 6.4.67 software. ENc designates the degree of codon bias for genes by computing deviation from uniform codon usage without any prior dependency over the sequence length or specific information of preferred codons [84]. 

## 5. Conclusions

In this study, we reported the first complete mitochondrial genome of *Potamotrygon leopoldi*, an endemic species from the Xingu River in the Amazon region. The mitogenome has a length of 17,504 bp, comprising 13 protein-coding genes (PCGs), 22 transfer RNAs (tRNAs), and 2 ribosomal RNAs (rRNAs), exhibiting a structure similar to that of other *Potamotrygon* species. Phylogenetic analysis revealed a sister relationship with *P. orbignyi* and *P. falkneri*, enhancing our understanding of Potamotrygonidae evolution. The mitogenomic data generated herein offer a foundational resource that can be adapted for future investigations into the migration patterns and adaptive mechanisms of stingrays. Specifically, these findings contribute valuable molecular insights for phylogenetic analyses, DNA barcoding, and metabarcoding, supporting a deeper exploration of genetic diversity and evolutionary trajectories in these ecologically and evolutionarily significant species.

## Figures and Tables

**Figure 1 ijms-26-08252-f001:**
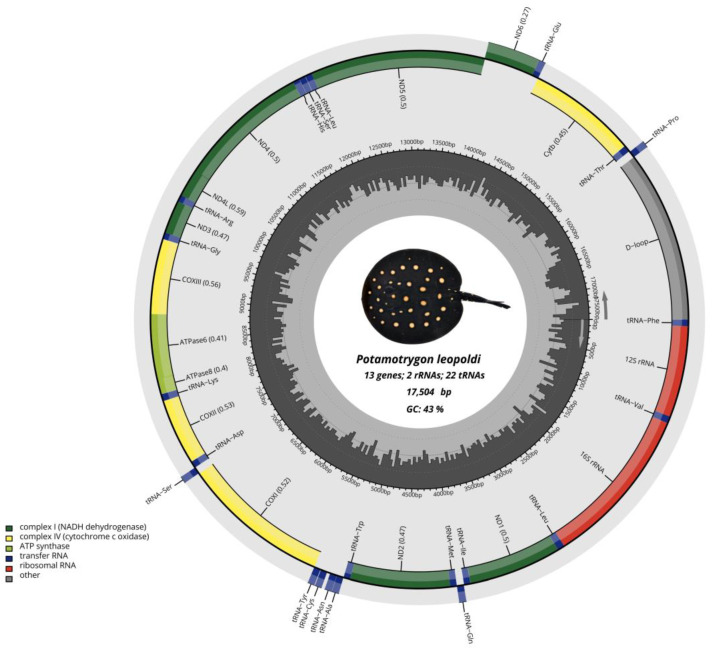
Circular genome map of the *P. leopoldi* mitochondrial genome, showing 13 PCGs (in green and yellow), 2 rRNA (in red) genes, 22 tRNA genes (in blue), and the control region D-loop (in gray).

**Figure 2 ijms-26-08252-f002:**
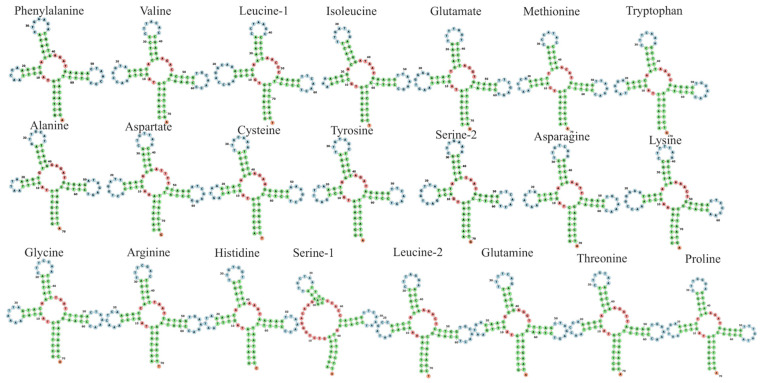
Secondary structure of tRNAs in the mitogenome of *P. leopoldi* visualized in the Forna web server.

**Figure 3 ijms-26-08252-f003:**
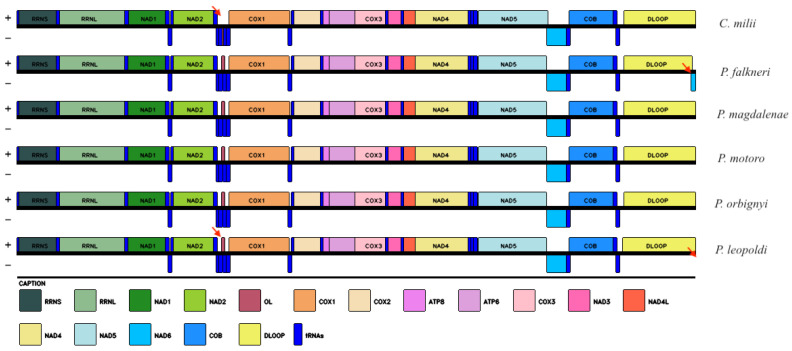
Arrangement of genes encoding rRNAs and proteins of *P. leopoldi* and other species of *Potamotrygon* genus. Heavy (+) and light (−) strands. Red arrows represent the changes between Potamotrygon leopoldi and the other species that were used in the gene arrangement comparison.

**Figure 4 ijms-26-08252-f004:**
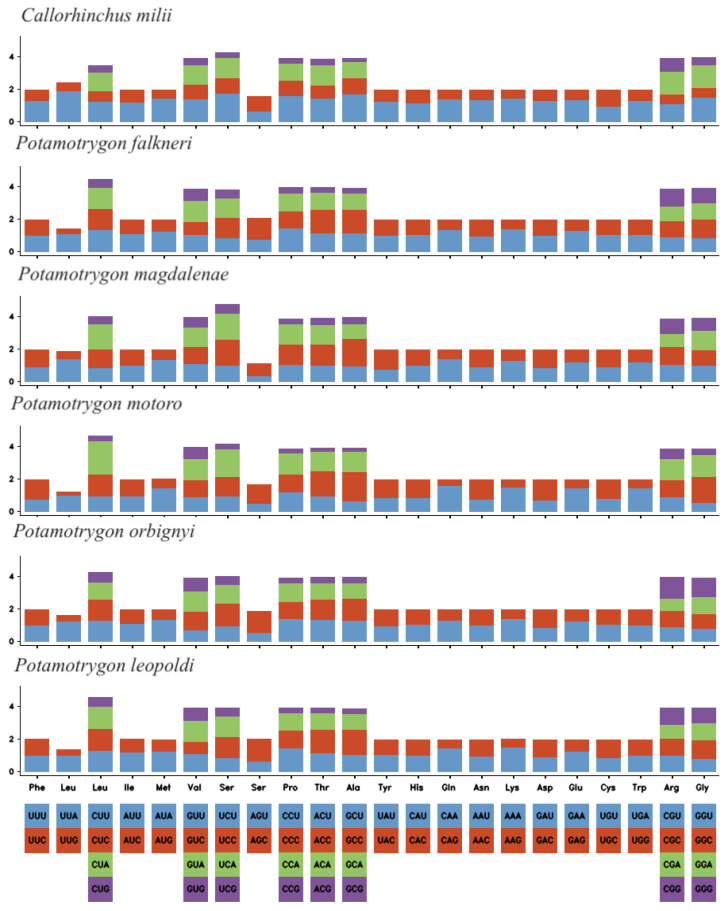
Relative Synonymous Codon Usage (RSCU) of the protein-coding genes of *P. leopoldi* and species from the Potamotrygonidae family and the *C. milii* as outgroup.

**Figure 5 ijms-26-08252-f005:**
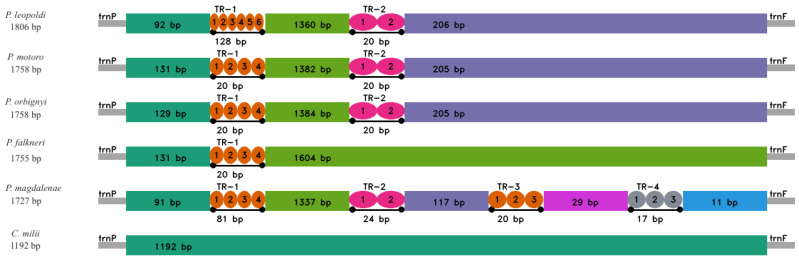
Organization of the CR in the complete mitogenomes of *P. leopoldi*, other four species from the *Potamotrygon* genus, and the outgroup *C. milii*.

**Figure 6 ijms-26-08252-f006:**
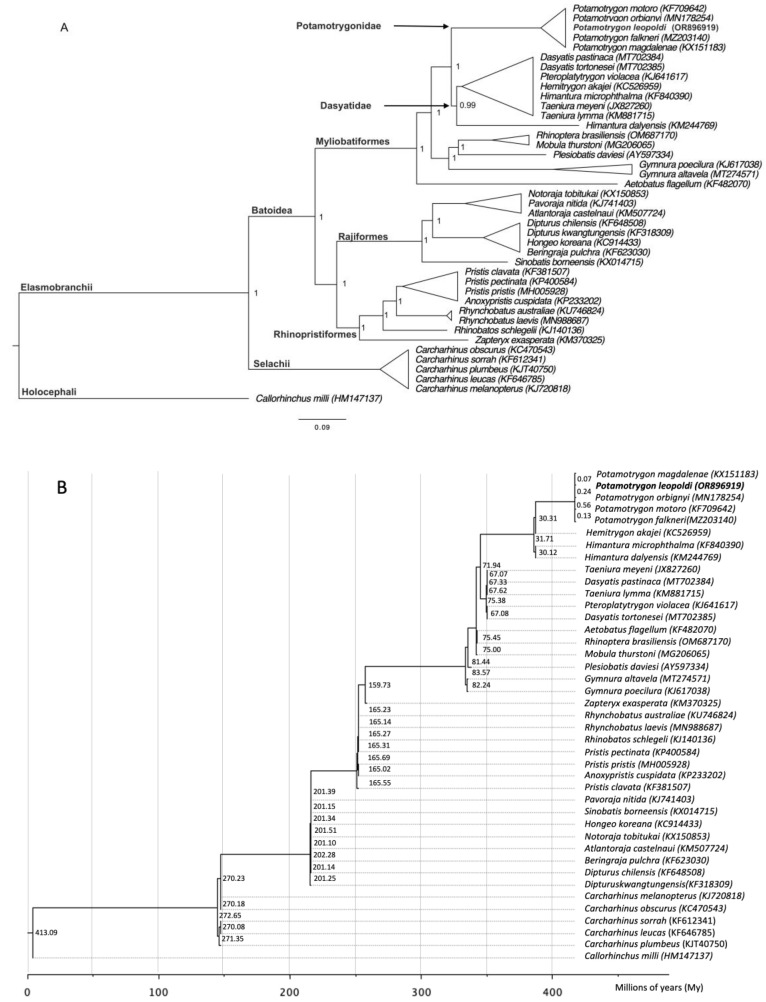
(**A**) Phylogenetic tree using the complete mitochondrial genome of species of order Myliobatiformes. GenBank Accession IDs are in brackets. The nodal support indicates the posterior probability in BI. (**B**) Divergence time estimates in the class Chondrichthyes. The value above the node represents the time to present in My.

**Table 1 ijms-26-08252-t001:** Mitochondrial genome organization of *Potamotrygon leopoldi*.

Name	Type	Strand	Start	Stop	Intergenic Nucleotide	Length	Anticodon and Start Codon/Stop Codon
*tRNA-Phe*	tRNA	+	1	69	0	69	-
*12S rRNA*	rRNA	+	70	1030	0	961	-
*tRNA-Val*	tRNA	+	1031	1102	19	72	-
*16S rRNA*	rRNA	+	1122	2788	3	1667	-
*tRNA-Leu*	tRNA	+	2792	2866	0	75	-
*ND1*	PCG	+	2867	3841	1	975	ATG/TAA
*tRNA-Ile*	tRNA	+	3843	3910	4	68	-
*tRNA-Gln*	tRNA	−	3915	3986	0	72	-
*tRNA-Met*	tRNA	+	3987	4056	0	70	-
*ND2*	PCG	+	4057	5103	−1	1047	ATG/TAA
*tRNA-Trp*	tRNA	+	5103	5172	1	70	-
*tRNA-Ala*	tRNA	−	5174	5243	1	70	-
*tRNA-Asn*	tRNA	−	5245	5317	4	73	-
*tRNA-Cys*	tRNA	−	5351	5419	5	69	-
*tRNA-Tyr*	tRNA	−	5425	5493	1	69	-
*COX1*	PCG	+	5495	7048	6	1554	GTG/TAG
*tRNA-Ser 2*	tRNA	−	7055	7125	0	71	-
*tRNA-Asp*	tRNA	+	7126	7196	3	71	-
*COX2*	PCG	+	7200	7890	0	691	ATG/T(AA)
*tRNA-Lys*	tRNA	+	7891	7964	1	74	-
*ATP8*	PCG	+	7966	8133	−10	168	ATG/TAA
*ATP6*	PCG	+	8124	8807	−1	683	ATG/TAA
*COX3*	PCG	+	8807	9592	5	786	ATG/TAA
*tRNA-Gly*	tRNA	+	9598	9668	0	71	-
*ND3*	PCG	+	9669	10,019	−1	351	ATG/TAA
*tRNA-Arg*	tRNA	+	10,019	10,089	0	71	-
*ND4L*	PCG	+	10,090	10,386	−7	297	ATG/TAA
*ND4*	PCG	+	10,380	11,760	3	1381	ATG/T(AA)
*tRNA-His*	tRNA	+	11,764	11,832	1	69	-
*tRNA-Ser*	tRNA	+	11,834	11,900	0	72	-
*tRNA-Leu*	tRNA	+	11,903	11,974	0	72	-
*ND5*	PCG	+	11,975	13,810	−4	1836	ATG/TAA
*ND6*	PCG	−	13,807	14,328	0	522	ATG/TAG
*tRNA-Glu*	tRNA	−	14,319	14,397	1	69	-
*CYTB*	PCG	+	14,399	15,541	4	1143	ATG/TAA
*tRNA-Thr*	tRNA	+	15,546	15,616	8	71	-
*tRNA-Pro*	tRNA	−	15,652	15,694	0	70	-
*D-loop*	CR		15,694	17,504		1810	-

## Data Availability

The mitogenome sequence data that support the findings of this study are openly available in GenBank of NCBI at https://www.ncbi.nlm.nih.gov (accessed on 16 August 2024), under accession number OR896919.

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
