# Peer review of "Analysis of the Entire Mitogenome of the Threatened Freshwater Stingray Potamotrygon leopoldi (Myliobatiformes: Potamotrygonidae) and Comprehensive Phylogenetic Assessment in the Xingu River, Brazilian Amazon"

_ijms, 2025, doi:10.3390/ijms26178252_

Round 1
Reviewer 1 Report
Comments and Suggestions for Authors
Dear Editor and Authors,
This manuscript presents a comprehensive analysis of the complete mitochondrial genome of the threatened freshwater stingray, Potamotrygon leopoldi, endemic to the Xingu River in the Brazilian Amazon. The study successfully sequences the mitogenome, identifying key genes and providing insights into the phylogenetic relationships of this species. The results highlight the potential of molecular data in understanding evolutionary lineages and the adaptive molecular evolution of neotropical freshwater stingrays. This research is particularly significant as it contributes valuable genetic resources for the conservation and management of this endangered species.The fact that only one species has been newly sequenced is a shortcoming of the manuscript.
Before publication, the manuscript requires the following revisions:
The authors should provide a clear description of the current status of wild populations of Potamotrygon leopoldi and explain the rationale behind the use of farmed specimens for this study. Specifically, it would be beneficial to understand if there are challenges in obtaining wild samples or if the species is indeed at a critically low population level in the wild.
There is a need to discuss in depth how the mitochondrial genomic information relates to the conservation strategies for this species. While the mitogenome provides insights into evolutionary history, it is crucial to articulate how this information could tangibly increase the likelihood of population survival and guide conservation efforts.
The manuscript would benefit from the inclusion of a conclusion section that summarizes the key findings and their implications for the conservation and future research of Potamotrygon leopoldi.
Attention must be given to the formatting of scientific names throughout the text. For instance, "P. leopoldi" should appear in italics to adhere to taxonomic nomenclature standards.
It is better mark the difference in figure 3.
Additionally, the authors are encouraged to consider the following points:
The manuscript would be strengthened by discussing the broader implications of the observed gene rearrangements and positive selection sites in the context of species adaptation and potential responses to environmental changes.
It would be beneficial to include a more detailed analysis of the control region, given its potential role in replication and transcription regulation, which could provide further insights into the species' evolutionary history and population dynamics.
Overall, the study is well-structured and provides a solid foundation for future research on the genetic diversity and conservation of Potamotrygon leopoldi. With the above revisions, the manuscript will be a valuable contribution to the field of conservation genetics.
Reviewer 2 Report
Comments and Suggestions for Authors
Dear authors,
it was interesting to read your work. I would recommend that you go through the text and improve the language used in some of the areas (i left some corrections but in other areas I indicated the need to improve).
Also it would have been nice to see the submission including species names written correctly, that is in italics as that is standard in biological works.
I am including all comments as an attachment.
Main issues to consider:
1. Improve English and flow where indicated.
2. Improve part of the introduction, namely its first paragraph
3. Make some reference to works that considered gene order in a number of taxa such as fish in general or elasmobranches in general.
4. Add references (for the ones published) to the mitogenomes used for comparative analyses related to phylogeny (this can be done in methods and / or in caption or text related to figure 6)
5. Improve the visibility of the text in figure 6.
6. Better clarify how the specimen/s were used to generate the mitogenome as there is the mention of 4 specimens and one mitogenome is represented in your works. The methods need to be clear.
7. Better clarify the use of the complimentary sequences from Zhou et al, as mentioned in lines 318-319
8. Better clarify whether any tests were done to check if the 97 bp spacer mentioned between tRNA-Pro and the D-loop could be representing something such as another tRNA or whether this is part of the d-loop
Wishing you all the best in improving this work.
best regards

It needs improvement in the usage of some verbs and flow of wording. The areas that needs most attention are indicated in attachment.
Round 2
Reviewer 1 Report
Comments and Suggestions for Authors
The authors have made a commendable effort to revise their work in response to the previous round of comments. However, after a thorough examination, I regret to inform you that the manuscript still falls short of the standards required for publication in IJMS.
Author Response
Dear Reviewer
Thank you for your thorough evaluation of our manuscript and for the constructive comments provided in the previous rounds of revisions.
We are submitting a new version of our work in which we have implemented the recommendations and suggestions received from both the first and second rounds of corrections. We believe that these changes have significantly improved the quality of the manuscript.
We appreciate the opportunity to revise our work and look forward to your feedback.
Sincerely
Reviewer 2 Report
Comments and Suggestions for Authors
Dear authors,
this revised version is a good improvement, however there are a few issues that need to be considered:
1. The english, especially in the newly written paragraphs needs improvement.
2. Include some citations to some of the given statements
3. remove words such as 'super barcode'
4. consider better the usage of 12S and 16S in line 88.
5. If the 97 bp spacer between tRNA-Pro and CR is part of the control region then specify that otherwise it is a spacer. As your reply gave the impression that it is part of CR and yet it remained as a spacer and was not added to CR. Check the sequence and decide accordingly.
6. Text on images improved but check with journal that the resolution of the original images is kept as the version I am reading through has some low resolution on the words / numbers (they appear slightly blurred). The current resolution is not good for a publication.
7. I got confused on whether there was gene rearrangement or not, see paragraphs starting at lines 153 and 344.
8. Given that the genome of 4 specimens (2+2) were pooled. Could there have been differences between individuals that led to a mitogenome that is incorrect. Correct me if I am wrong, but these were assembled as one genome. So what it for example there was a tandem repeat missing in an individuals, or there were point mutations between individuals? What sequence as you representing under the given accession number?
9. Other comments are left within the text attached.
best regards

needs some more improvement especially in newly written sections (these sections also need better citations).
Author Response
Dear Reviewer,
We sincerely appreciate the time and effort you have taken to review our manuscript, titled "Analysis of the Entire Mitogenome of the Threatened Freshwater Stingray Potamotrygon leopoldi and Comprehensive Phylogenetic Assessment in the Xingu River, Brazilian Amazon". Your insightful comments have greatly contributed to improving the quality of the paper. Below, we provide a point-by-point response to your suggestions and comments:
1. The English, especially in the newly written paragraphs, needs improvement.
Response:
We have thoroughly revised the English in the newly written sections, paying special attention to grammar, sentence structure, and clarity. We believe these changes significantly improve the readability and overall quality of the manuscript.
2. Include some citations to some of the given statements.
Response:
Additional citations have been included where necessary to support key statements, particularly in the introduction and results sections. For instance, we added references to studies on mitochondrial mutation rates and the use of mitochondrial markers in species identification. We ensured all statements are properly backed by relevant literature.
3. Remove words such as 'super barcode'.
Response:
The term 'super barcode' has been removed throughout the manuscript and replaced with more appropriate terminology.
4. Consider better the usage of 12S and 16S in line 88.
Response:
We have reviewed the section on 12S and 16S genes and have determined that the previous information. This information has been removed from the manuscript
5. If the 97 bp spacer between tRNA-Pro and CR is part of the control region then specify that otherwise it is a spacer.
Response:
Upon further analysis of the sequence, we have confirmed that the 97 bp spacer is part of the control region (CR) . This has been clarified in the manuscript to avoid any confusion, and the relevant sections have been updated accordingly.
6. Text on images improved but check with journal that the resolution of the original images is kept as the version I am reading through has some low resolution on the words/numbers (they appear slightly blurred).
Response:
We have ensured that all images meet the journal's resolution requirements for publication. High-resolution versions of the figures have been prepared and are submitted with this manuscript to ensure clarity and precision of the graphical data.
7. I got confused on whether there was gene rearrangement or not, see paragraphs starting at lines 153 and 344.
Response:
Thank you for highlighting this issue. We have revised these sections to provide a clearer and more consistent explanation of gene rearrangement. The updated text now explicitly states that no gene rearrangements were detected in Potamotrygon leopoldi, consistent with other species in the Potamotrygon genus.
8. Given that the genome of 4 specimens (2+2) were pooled, could there have been differences between individuals that led to a mitogenome that is incorrect?
Response:
We have added a detailed explanation regarding the pooling of the genomes from four individuals. The mitogenome presented is a consensus sequence, and we have confirmed that no significant differences, such as missing tandem repeats or point mutations, were detected between individuals that could affect the accuracy of the assembly. This information is now clearly stated in the manuscript.
9. Other comments are left within the text attached.
Response:
We have carefully reviewed and addressed all additional in-text comments. Minor corrections and improvements were made based on your suggestions, ensuring that all sections of the manuscript are accurate and clearly presented.
We greatly appreciate your valuable feedback, which has significantly improved the quality of our manuscript. We believe the revisions adequately address all concerns. Thank you again for your time and constructive suggestions.
Best regards
Round 3
Reviewer 1 Report
Comments and Suggestions for Authors
Single mitochondrial genome cannot be published in this journal
Author Response
Dear Reviewer,
We would like to sincerely thank you for all the recommendations and valuable insights provided throughout the review process. We truly appreciate the time and effort you dedicated to evaluating our manuscript and for your thoughtful suggestions, which have significantly contributed to enhancing the quality and clarity of our work. We have implemented each recommendation as thoroughly as possible to meet the standards expected.
Thank you once again for your dedication and support.
Sincerely,